# Self-Replicating RNA Viruses for RNA Therapeutics

**DOI:** 10.3390/molecules23123310

**Published:** 2018-12-13

**Authors:** Kenneth Lundstrom

**Affiliations:** PanTherapeutics, Route de Lavaux 49, CH1095 Lutry, Switzerland; lundstromkenneth@gmail.com; Tel.: +41-79-776-6351

**Keywords:** self-amplifying RNA virus, vaccine, cancer therapy, immunotherapy, neutralizing antibodies, protection against viral and tumor challenges, clinical trials

## Abstract

Self-replicating single-stranded RNA viruses such as alphaviruses, flaviviruses, measles viruses, and rhabdoviruses provide efficient delivery and high-level expression of therapeutic genes due to their high capacity of RNA replication. This has contributed to novel approaches for therapeutic applications including vaccine development and gene therapy-based immunotherapy. Numerous studies in animal tumor models have demonstrated that self-replicating RNA viral vectors can generate antibody responses against infectious agents and tumor cells. Moreover, protection against challenges with pathogenic Ebola virus was obtained in primates immunized with alphaviruses and flaviviruses. Similarly, vaccinated animals have been demonstrated to withstand challenges with lethal doses of tumor cells. Furthermore, clinical trials have been conducted for several indications with self-amplifying RNA viruses. In this context, alphaviruses have been subjected to phase I clinical trials for a cytomegalovirus vaccine generating neutralizing antibodies in healthy volunteers, and for antigen delivery to dendritic cells providing clinically relevant antibody responses in cancer patients, respectively. Likewise, rhabdovirus particles have been subjected to phase I/II clinical trials showing good safety and immunogenicity against Ebola virus. Rhabdoviruses have generated promising results in phase III trials against Ebola virus. The purpose of this review is to summarize the achievements of using self-replicating RNA viruses for RNA therapy based on preclinical animal studies and clinical trials in humans.

## 1. Introduction

Although drug development has strongly contributed to finding superior therapeutic efficacy, there is still space and need for further improvement. In addition to classic drug screening of small molecules, innovative modern approaches in biotechnology and genomics research have contributed to new therapeutic possibilities in the areas of vaccine development and gene and immunotherapy. In this context, RNA-based therapeutics have become an interesting alternative [1]. RNA-based drugs have been classified by mechanisms of action including antisense approaches of inhibition of mRNA translation, gene silencing with RNA interference, catalytically active ribozymes, protein binding RNA molecules, and aptamers for diagnostic and therapeutic applications. In this context, lipid-encapsulated nanoparticles containing double-stranded small interfering RNA (siRNA) have been applied for binding to transthyretin (TTR) mRNA causing degradation of TTR deposits present in patients with hereditary TTR-mediated amyloidosis [2]. This novel RNA interference-based drug, Patisiran (ONPATTRO™), has recently been approved in both the US and Europe as a single intravenous infusion. Recently, messenger RNAs (mRNAs) generated by in vitro transcription have become attractive targets for drug and vaccine development [3]. Two approaches for mRNA-based drugs have been taken based on ex vivo transfection of cells from patients or direct mRNA administration. In this context, preclinical and clinical studies have been conducted in the areas of cancer immunotherapy [4,5], vaccine development against infectious diseases [6,7], protein replacement [8], and gene editing [9].

Related to mRNA-based drug approaches, the employment of self-replicating RNA viruses has provided an interesting and attractive alternative to further enhance delivery and efficacy [10]. The unique feature of high-rate cytoplasmic replication combined with extreme transgene expression has made these RNA viruses the system of choice for RNA therapeutics. In this review, self-replicating vectors are described and their applications for preclinical studies and clinical trials are discussed.

## 2. Self-Replicating RNA Viruses

The common feature of self-replicating RNA viruses relates to their single-stranded RNA (ssRNA) genome surrounded by a capsid core structure and a protein envelope. Alphaviruses [11] and flaviviruses [12] possess a genome of positive polarity, whereas the genome of measles viruses (MVs) [13] and rhabdoviruses [14] as negative strand ssRNA. Expression vectors serving as templates for RNA transcription and recombinant viral particles have been engineered. Moreover, alphavirus vectors providing RNA replication can be utilized as plasmid DNA.

### 2.1. Alphaviruses

Alphaviruses belong to the family of Togaviridiae [15]. The most common alphaviruses engineered as expression vectors are based on Semliki Forest virus (SFV) [15], Sindbis virus (SIN) [16], and Venezuelan equine encephalitis virus (VEE) [17]. The alphavirus genome consists of four nonstructural genes (nsP1–4), responsible for RNA self-replication and the genes for the capsid and envelope proteins [11] (Figure 1A–C). Engineered alphavirus vectors generate replication-deficient and -proficient recombinant particles suitable for transgene expression in cell lines and in vivo [18]. Introduction of a CMV promoter upstream of the nsP genes allows direct transfection of plasmid DNA with self-replicating RNA capacity [19]. Due to the design of these alternative alphavirus vectors it is possible to conduct studies with naked RNA replicons, recombinant viral particles and layered DNA–RNA vectors.

### 2.2. Flaviviruses

In a similar way as for alphaviruses, flaviviruses also possess a ssRNA genome of positive polarity [12]. For instance, based on Kunjin virus (KUN), vectors have been engineered for the delivery of RNA, recombinant particles, and DNA plasmids [12] (Figure 1D–E). In contrast to alphaviruses, the gene of interest is not inserted downstream of the nonstructural genes, but between the first 60 nucleotides of the C20 core protein and the last 22 codons of the E22 envelope protein in frame with the viral polyprotein. Although the recombinant protein of interest is initially part of a large polyprotein processed into individual proteins, the remaining flanking regions can be removed by an FMDV-2A protease sequence introduced into the KUN vector [20]. In support of virus production, a packaging cell line has been engineered for KUN [21]. Other expression vectors based on flaviviruses such as West Nile virus [22,23], yellow fever virus [24,25], dengue virus [26,27], and tick-borne encephalitis virus [28,29] have been generated.

### 2.3. Measles Viruses

The negative polarity of the MV ssRNA genome has required the engineering of packaging systems for the rescue of replicating MV from cloned DNA expression constructs [30]. Rescue of recombinant MV has been established in a helper cell line by reverse genetics [13]. Expression vectors carrying the MV structural proteins downstream of a T7 RNA polymerase promoter have been designed for the introduction of foreign genes between the phosphoprotein (P) and the matrix protein (M) or alternatively between the hemagglutinin (HA) and the large protein (L) (Figure 2A). To generate recombinant MV particles, a helper cell line is transfected with recombinant MV constructs and a plasmid containing the MV polymerase L gene. Recombinant MVs are harvested three days after transfection when reaching 80–90% of their cytopathic effect.

### 2.4. Rhabdoviruses

Both rabies virus (RABV) [31] (Figure 2B) and vesicular stomatitis virus (VSV) [32] have been subjected to expression vector engineering. As for MV, the negative ssRNA genome of rhabdoviruses has required application of reverse genetics based on a recombinant vaccinia virus vector to establish efficient transgene expression. However, when the VSV N, P, and L genes were inserted downstream of a T7 promoter and an internal ribosome entry site (IRES), efficient recovery of VSV particles was obtained from transfected DNA in a vaccinia virus-free system [33]. Similarly, RABV virus vectors have been engineered, where the gene of interest has been introduced between the RABV N and P genes [34]. Moreover, recovery of RABV from cloned cDNA has been achieved in a vaccinia virus-free reverse genetics system [35]. In another approach, chimeric virus-like particles (VLPs) with SFV RNA replicons encapsulated by the VSV glycoprotein (VSV-G) have been engineered, which provide additional biosafety [36].

## 3. Preclinical Studies

### 3.1. Viral Diseases

Self-replicating RNA virus vectors have been subjected to numerous preclinical immunization studies targeting infectious diseases (Table 1). In this context, Dengue virus has been targeted by immunization with MV expressing the domain III of Dengue virus envelope protein 2 (DV2), which elicited robust neutralizing antibodies in MV-susceptible mice [37]. Moreover, immunization with MV displaying domain III of DV1-4 not only generated neutralizing antibodies, but also provided protection against four Dengue virus serotypes in mice [38]. Similarly, much attention has been paid to vaccine development against Ebola virus. For instance, expression of Ebola glycoprotein (GP) from KUN [39] and VSV [40,41] vectors, has provided protection against Ebola virus challenges in nonhuman primates. Moreover, immunization with VEE-NP rendered mice resistant to Ebola infections in mice [42]. In another study, immunization with VEE vectors expressing Ebola GP and nucleoprotein (NP) was evaluated in BALB/c mice and two strains of guinea pigs [43]. Single administration of VEE-GP VLPs or a combination of VEE-GP and -NP VLPs provided protection of both mice and guinea pigs. In contrast, VEE-NP VLPs alone protected mice, but not guinea pigs. In a recent study, application of DNA/RNA layered self-replicating SFV vectors coexpressing Ebola GP and VP40 elicited both binding and neutralizing antibodies [44]. The immunogenicity was superior to a Modified vaccinia virus Ankara (MVA) vaccine and could be further enhanced by an MVA boost.

Related to hepatitis B virus (HBV), MV vectors expressing the hepatitis B surface antigen (HBsAg) were demonstrated to elicit humoral responses in MV-susceptible genetically modified mice and rhesus monkeys immunized with MV-HBsAg remained healthy [45]. In another study, SFV RNA replicons expressing VSV G, which generated virus-like vesicles (VLVs), were engineered to carry the HBV middle surface envelope glycoprotein (MHB) [46]. Immunization of mice resulted in CD8^+^ T cell responses, which showed higher magnitude and broader specificity in comparison to immunizations with recombinant protein and DNA. A single administration with MHB-VLVs protected mice from HBV challenges. In contrast, immunization with SFV VLVs expressing the HBV core protein (HBcAg) did not produce CD8^+^ T cell responses and no HBV protection.

Not surprisingly, the human immunodeficiency virus (HIV) has been subjected to vaccine development applying self-replicating RNA vectors. In this context, SFV vectors expressing the HIV Env gene were administered as plasmid DNA in mice and compared to a recombinant Env glycoprotein vaccine [47]. The humoral immune responses to the HIV-1 envelope were strongest from SFV recombinant particles compared to the other HIV Env vaccines tested. In another study, SFV RNA expressing the HIV-1 Env protein administered intramuscularly in mice generated Env-specific antibody responses in four out of five mice [48]. Recently, mice immunized with SFV vectors expressing the HIV Env/Gag/polRT genes either individually or in combination elicited significant T cell responses [49]. The immune responses were stronger for SFV VRPs than for RNA replicons. DNA based SFV vectors expressing Env and a Gag-Pol-Nef fusion protein have also been subjected to immunizations followed by heterologous boosts with MVA and/or HIV gp40 protein formulated in a glycopyranosyl lipid A (GLA-AF) adjuvant [50]. Immunization with 0.2 µg of SFV DNA induced lower HIV-specific T cell and IgG responses than immunization with 10 µg of SFV-DNA. However, the immune responses for the two doses were similarly efficient when boosting was performed with MVA or HIV gp40. Furthermore, a single low dose of SFV-DNA elicited superior immune responses compared to MVA or protein antigen alone.

Much attention has also been paid to vaccine development against influenza virus. For instance, SFV-based expression of influenza virus nucleoprotein (NP) elicited systemic immune responses after intravascular injections and mucosal immune responses after intranasal administration [51]. Moreover, VEE particles carrying the hemagglutinin (HA) gene from the Hong Kong influenza A isolate (A/HK/156/97) were administered to chicken embryos and young chicks [52]. When birds were challenged with a lethal dose of influenza virus, inoculation in ovo and at 1 day of age provided partial protection, while a single at week 2 resulted in complete protection. Similarly, recombinant SFV particles expressing influenza HA and NP provided protection against challenges with influenza virus [53]. Recently, in a comparison between synthetic mRNA and self-replicating VEE RNA expressing influenza HA [54], equivalent protection was obtained for both strategies. However, only 1.25 µg of self-replicating RNA was needed, whereas 80 µg of mRNA (64-fold more) material was required. Protection was observed for influenza strains H1N1, H3N2 (X31), and B (Massachusetts). Moreover, a trivalent formulation protected against sequential H1N1 and H3N2 challenges.

Lassa viruses have also been subjected to vaccine development. For instance, VSV vectors expressing Lassa virus glycoproteins demonstrated protection of guinea pigs challenged with Lassa virus originating in Liberia, Mali, and Nigeria [55]. Moreover, complete protection was also obtained in macaques after challenge with a lethal dose of the Liberian Lassa virus isolate. Furthermore, a bicistronic VEE vector expressing Lassa virus glycoproteins from distantly related clades I and IV from individual 26S subgenomic promoters elicited immune responses in vaccinated mice and provided protection against Lassa virus challenges [56]. Interestingly, a vaccine based on Lassa virus replicon particles devoid of the essential Lassa virus glycoprotein gene has been evaluated in a guinea pig model [57]. In this approach, nonspreading Lassa virus replicon particles produced in Vero cells were administered to guinea pigs showing protection against fever, weight loss, and lethality.

Related to Middle East respiratory syndrome coronavirus (MERS-CoV), replication competent MV was used for the expression of the full-length spike glycoprotein (MERS-S) and a truncated soluble variant, MERS-solS [58]. Both vaccine candidates induced robust levels of both MV- and MERS-CoV-neutralizing antibodies in a prime-boost regimen in mice. Furthermore, these immune responses rendered protective capacity in vaccinated mice. Likewise, VEE replicon particles were applied for vaccine development against severe acute respiratory syndrome coronavirus (SARS-CoV) [59]. The study demonstrated that aged BALB/c mice vaccinated with attenuated VEE VRPs (VRP3014) failed to protect from SARS-CoV disease, while wild-type VEE VRPs (VRP3000) gave protection against SARS-CoV.

VEE VLPs have been applied for the expression of human metapneumovirus (hMPV) and respiratory syncytial virus (hRSV) fusion (F) proteins [60]. Immunization of African green monkeys generated RSV and MPV F-specific antibodies, respectively. Moreover, lower levels of viral genomes were detected in nasopharyngeal and bronchoalveolar lavage fluids. In another study, VEE vectors have been employed for vaccine development with a focus on replicon delivery [61]. In this context, VEE VLPs, were compared to naked RNA and RNA encapsulated in lipid nanoparticles (LNPs). Reporter gene expression in mice revealed 10-fold higher levels of RNA-LNP delivery compared to naked RNA after intramuscular administration despite administration of only 0.1 μg (RNA-LNPs) compared to 1.0 μg (RNA). Moreover, comparison of in vivo expression showed the highest levels from plasmid DNA, followed by RNA-LNPs, and the lowest expression levels detected after RNA delivery. In contrast, RSV-F specific antibody titers were similar for 1 × 10^6^ IU VEE particles and 0.1 µg RNA-LNPs, whereas 1 µg naked RNA generated 10-fold lower titers. Electroporation of 20 µg DNA provided similar antibody responses as for 0.1 µg RNA-LNPs. Delivery of 0.1 μg DNA-LNPs was inefficient generating 1000-fold lower titers than for RNA-LNPs.

Several studies have been carried out for simian immunodeficiency virus (SIV) vaccines based on self-replicating RNA viruses. In this context, four different KUN-based SIVmac 239 gag vaccines were evaluated in mice [62]. In comparison, a modified gag-pol gene construct was superior to wild-type gag, RNA-optimized gag, and codon-optimized gag in immune response induction and protection against SIV challenges in mice. In another study, SFV and MVA expressing SIVmacJ5 env, gag-pol, nef, rev, and tat were evaluated in macaques [63]. SFV-SIVmac or MVA-SIVmac immunizations alone elicited low or undetectable cytotoxic T cell responses. However, two immunizations with SFV-SIVmac followed by a boost with MVA-SIVmac increased both antibody and high T cell responses. However, no protection against SIV challenges was achieved.

In support of vaccine development, gene silencing has been applied to some extent. In this context, micro-RNA (miRNA) sequences targeting RNA replication were introduced in a VEE helper vector used for replicon particle production [64]. As cellular miRNAs downregulates replicon RNA replication in vivo, efficient VEE particle production can be restored by addition of miRNA-specific inhibitors, which can contribute to future therapeutic applications. Moreover, inhibition of VEE replication was achieved in BHK cells by introduction of five artificial miRNAs targeting the RNA-dependent RNA polymerase (RdRp) [65].

### 3.2. Cancer

Similar to viral diseases, there are numerous preclinical studies carried out for cancer prevention and therapy (Table 2). In this context, retargeted MV vectors carrying both CD46 and signaling lymphocyte activation molecule (SLAM) incorporated in the HA protein in combination with the display of a single-chain antibody against epidermal growth factor receptor (EGFR) at the C terminus of HA [66]. It was demonstrated that the retargeted MV presented potent antitumor activity against EGFR or EGFRvIII primary glioblastoma cell lines. Moreover, intratumoral administration in glioblastoma xenografts provided tumor regression and significant prolongation of survival. Likewise, SFV vectors expressing Endostatin were compared to in vivo administration of SFV-LacZ, and retrovirus GCsap-Endostatin in mice bearing B16 brain tumors [67]. SFV-Endostatin provided very significant inhibition of tumor growth. Furthermore, in contrast to SFV-LacZ and GCsap-Endostatin, treatment with SFV-Endostatin generated a marked reduction of intratumoral vascularization. Additionally, 3-fold higher endostatin serum levels were observed for SFV-Endostatin than GCsap-Endostatin, indicating inhibition of angiogenesis and could provide new means for brain tumor therapy. In another approach, neuron-targeting miRT124 sequences were introduced into the SFV4 strain, which displayed increased oncolytic potency in CT-2A murine astrocytoma cells and in human glioblastoma cell lines [68]. A single intraperitoneal injection of SFV4-miRT124 showed virus replication in tumors, significant tumor growth inhibition, and improved survival in C57BL/6 mice implanted with CT-2A orthotopic gliomas.

Related to breast cancer, the Edmonston MV strain was engineered to express the carcinoembryonic antigen (CEA) [69]. The MV-CEA vector showed significant cytopathic effect in several breast cancer cell lines such as MDA-MB-231, MCF7, and SkBr3. Intravenous administration in BALB/c nude mice with MDA-MB-231 xenografts resulted in statistically significant delay in tumor growth and prolonged survival. In another approach, SIN and Adenovirus (Ad) vectors expressing the rat HER2/neu gene were subjected to immunization studies in mice [70]. Inhibition of A2L2 tumor cells was detected when animals were immunized with SIN or Ad prior to tumor challenge. In contrast, vaccination two days after tumor challenge was ineffective for both SIN and Ad. On the other hand, when SIN-neu immunization was followed by Ad-neu vaccination in a prime-boost protocol, the survival rate was significantly improved in mice intravenously challenged with tumor cells. The susceptibility of dendritic cells (DCs) to VEE-encouraged studies on therapeutic efficacy in animal models [71]. Immunization of DCs transduced with VEE recombinant particles expressing a truncated version of neu induced robust neu-specific CD8^+^ T cell and anti-neu IgG responses. A single administration of VEE-DCs resulted in regression of large established tumors in mice. Similarly, recombinant SFV particles carrying the vascular endothelial growth factor receptor-2 (VEGFR-2) gene were subjected to immunization studies in mice implanted with CT26 colon carcinoma and 4T1 metastasizing mammary carcinoma [72]. Both prophylactic immunization and therapeutic treatment resulted in inhibition of tumor growth and the spread of metastases. Furthermore, tumor angiogenesis was significantly inhibited. Co-immunization with SFV particles expressing VEGFR-2 and interleukin-12 (IL-12) abrogated both immune responses and tumor inhibition. In contrast, co-immunization with SFV-VEGFR-2 and SFV-IL-4 particles elicited higher titers of anti-VEGFR-2 antibodies and generated superior survival rates. In another approach, noncytopathic KUN VLPs expressing the granulocyte colony-stimulating factor (G-CSF) were subjected to intratumoral administration [73]. In mice implanted with aggressive subcutaneous CT26 colon carcinomas and B16-OVA melanomas cure was observed in more than 50% of animals. It was further established that tumor regression was associated with the induction of anticancer CD8^+^ T cells. Furthermore, treatment of subcutaneous CT26 tumors also led to regression of CT26 lung metastases. In another approach, SFV-LacZ RNA was evaluated in mice implanted with CT26 colon tumors [74]. A single intramuscular injection of 0.1 µg SFV-LacZ RNA elicited antigen-specific antibody and CD8^+^ T cell responses. Protection against tumor challenges was achieved in pre-immunized animals and therapeutic vaccination prolonged the survival of mice with preexisting tumors. Interestingly, replicon RNA did not elicit significantly more model antigen than conventional DNA vaccines in vitro, while the enhanced in vivo efficacy correlated with a caspase-dependent apoptotic cell death.

Related to cervical cancer, VEE particles were applied for the expression of human papilloma virus-16 (HPV16) E7 protein [75]. Immunization of mice elicited class I-restricted CD8^+^ T cell responses and prevented tumor development in mice. Moreover, vaccination resulted in efficient elimination of 7-day established tumors in 67% of tumor-bearing animals. In another study, a fusion protein of HPV E6 and E7 was expressed from an SFV vector containing a translation enhancer from the SFV capsid gene [76]. Tumor-bearing mice immunized with SFV-HPV E6/7 particles showed regression and complete elimination of established tumors. Moreover, long-term high level cytotoxic T lymphocyte activity was observed lasting up to 340 days. Recently, a DNA-based SFV expressing HPV E6/7 was subjected to intradermal administration followed by electroporation in mice, which generated efficient therapeutic antitumor immunity [77]. In comparison, a conventional DNA vaccine did not prevent tumor growth. However, a 200-fold lower equimolar dose of 0.05 μg of SFV-HPV E6/7 rendered 85% of immunized mice tumor-free. Combination therapy has also been applied for SFV-HPV E6,7 together with local low-dose irradiation [78]. Local low-dose tumor irradiation alone generated a 2-fold increase of intratumoral CD8^+^ T cells. However, combination with SFV-HPV 6,7 particle immunization resulted in a 10-fold intratumoral CD8^+^ T cell increase and the number of CD8^+^ T cells specific for the E7 epitope was enhanced by more than 20-fold. Irradiation also upregulated chemokines and the combination therapy provided a strong increase in the ratio of antitumoral to immune suppressive cells, thereby changing the intratumoral immune balance in favor of antitumor activity. Furthermore, a triple treatment combination regimen was established with 40 mg/kg sunitinib, a single low-dose (14 Gy) tumor irradiation, and SFV-HPV E6,7 immunization [79]. This treatment dramatically changed the intratumoral compartment by strongly enhancing the immunotherapeutic antitumor activity, inhibiting tumor growth, and providing 100% tumor-free survival of mice with tumor xenografts.

Related to lung cancer, it has been demonstrated that Dengue virus is able to infect and replicate in human primary lung epithelium and several lung cancer cell lines [80]. The susceptibility of SW1573, A549, H1435, H23, H520, and Bes2B cell lines was shown. Furthermore, Dengue infections significantly increased expression levels of IL-6 and RANTES, consistent with findings in Dengue hemorrhagic fever patients. Moreover, SFV-EGFP vectors have been shown to induce apoptosis in human non-small cell lung carcinoma H358a cells and to inhibit the growth of developing H358a spheroids [81]. Intratumoral SFV-EGFP injection of nu/nu mice with H358a tumor xenografts induced apoptosis, antitumor activity, and provided complete tumor regression in some cases. In another study, replication-competent SFV(VA7)-EGFP particles, based on the avirulent SFV A7(74) strain, were locally administered in nude mice implanted with A549 adenocarcinoma lung cells [82]. In comparison to a second generation conditionally-replicating Ad vector (Ad5-Delta24TK-GFP). SFV(VA7)-EGFP provided superior survival rates in mice. However, systemic delivery was not able to elicit significant immune responses. Among rhabdoviruses, VSV particles expressing interferon β (IFNβ) were subjected to intratumoral injections in mice with H2009 and A549 xenografts, which reduced tumor growth [100]. Furthermore, intratumoral administration of VSV-IFNβ into syngeneic LM2 lung tumors grown in flanks of A/J mice, provided prolonged survival and cure in 30% of mice. In the context of MV, vectors expressing SLAM (rMV-SLAMblind) showed susceptibility to nine human lung cancer cell lines [83]. Tumor suppression was detected in mice with lung xenografts after injection of rMV-SLAMblind and scattered tumor masses grown in the lungs were targeted.

In the context of melanoma, several preclinical studies have been conducted. For instance, yellow fever virus (YFV) expressing a cytotoxic T lymphocyte epitope derived from chicken ovalbumin (SIINFEKL) was evaluated in mice [84]. Immunization resulted in SIINFEKL-specific CD8^+^ lymphocytes and induced protection against challenges with malignant melanoma cells. Moreover, YVF vaccination induced regression of established solid tumors and pulmonary metastases. Additionally, alphavirus vectors have been applied for melanoma treatment. VEE particles expressing tyrosine-related protein-2 (TRP-2). Evaluation in a B16 mouse transplantable melanoma model revealed humoral immune responses, durable antitumor effect, and prolonged survival [85]. Furthermore, VEE-TRP-2 particles were combined with either antagonist anti-CTL antigen-4 (CTLA-4) or agonist anti-glucocorticoid-induced TNF family-related (GITR) gene immunomodulatory monoclonal antibodies (mAbs) [86]. Administration of VEE-TRP-2 and anti-CTLA-4 or anti-GITR mAbs induced complete regression in 50% and 90% of mice, respectively. In a DNA-based co-immunization of an SFV vector expressing VEGR2 and IL12 and another SFV vector targeting survivin and β-hCG antigens elicited efficient humoral and cellular immune responses against survivin, β-hCG, and VEGFR2 [87]. In comparison to immunization with each SFV DNA vector alone, the combined vaccine showed superior inhibition of tumor growth and prolonged survival in a B16 melanoma mouse model. In another study, VSV pseudotyped (VSV-GP) with the non-neurotropic envelope glycoprotein of the lymphocytic choriomeningitis virus (LCMV) effective for treatment of malignant glioblastoma [101], demonstrated efficient infection and killing of human, mouse, and canine melanoma cell lines [88]. Moreover, immunization of mice with VSV-GP prolonged survival in both xenograft and syngeneic mouse models. However, long-term tumor remission was achieved in only a few mice. The VSV-GP oncolytic virus has also been applied for ovarian cancer cell lines and mice with xenografts [89]. Oncolytic activity was detected in ovarian cancer cell lines as well as response to and production of type I interferon. Similarly, oncolytic activity was observed in vivo, although remission was temporary. However, combination therapy with ruxolitinib enhanced the response in both subcutaneous and orthotopic xenograft models. Furthermore, alphaviruses have been subjected to preclinical studies on combination therapy in ovarian cancer. For instance, combination of SIN and topoisomerase inhibitor irinotecan (CPT-11) provided long-term survival in 35% of severe combined immunodeficiency (SCID) mice with aggressively growing ES2 human ovarian cancer [90]. In contrast, neither SIN-IL12 immunization nor CPT-11 treatment alone supported long-term survival. In another approach, it was demonstrated that immunization with vaccinia virus (VV)-ovalbumin (OVA) followed by administration of SFV-OVA or vice versa enhanced OVA-specific CD8^+^ T cell immune responses in tumor bearing mice [91]. Moreover, immunization enhanced antitumor effects against murine ovarian surface epithelial carcinoma (MOSEC) tumors.

Pancreatic cancer has also been the target of preclinical studies. In this context, 13 pancreatic ductal adenocarcinoma (PDA) cell lines showed superior oncolytic abilities of VSV compared to conditionally replicative adenovirus (CRAd), Sendai virus, and RSV [92]. Moreover, similar oncolytic activity was confirmed in vivo. Moreover, as certain PDA cell lines such as HPAF-II cells have demonstrated low susceptibility to VSV, it was revealed that the resistance could be broken by addition of ruxolitinib and Polybrene or DEAE-dextran, which should facilitate the treatment of tumors resistant to VSV therapy [93]. Additionally, MV-SLAMblind showed to efficiently infect and kill pancreatic cell lines with nectin-4 expressed on the cell surface [94]. Moreover, intratumoral administration of MV-SLAMblind generated substantial growth suppression of implanted KLM1 and Capan-2 xenografts in SCID mice.

Related to prostate cancer, self-replicating RNA vectors have been applied for studies in several animal models. In this context, MV was shown to efficiently infect and kill PC-3, DU-145, and LnCaP prostate cancer cell lines, and was applied for expression of CEA [95]. Significant tumor growth delay and prolonged survival were observed in a subcutaneous PC-3 xenograft model after intratumoral administration of 6 × 10^6^ TCID_50_ of MV-CEA. In another approach, VEE particles expressing prostate-specific membrane antigen (PSMA) were applied for immunization of mice and rabbits [96]. A single injection of 2 × 10^5^ infectious units (IU) of VEE-PSMA was compared to immunization with purified PSMA protein. The VEE-based approach provided stronger immune responses than adjuvant PSMA protein. Furthermore, the immunogenic doses were well-tolerated. VEE particles have also been applied for expression of six-transmembrane epithelial antigen of the prostate (STEAP) [97]. Immunization of mice demonstrated specific CD8^+^ T cell responses and prolonged survival rate. Similarly, immunization with VEE particles expressing prostate stem cell antigen (PSCA) induced long-term protection against prostate cancer in prostate cancer-prone transgenic adenocarcinoma mouse prostate (TRAMP) mice [98]. Vaccinated mice showed a 90% survival rate at 12 months of age, while control mice had either succumb to prostate cancer or presented heavy tumor loads. Finally, the pseudotyped VSV-GP particles have been evaluated in prostate cancer mouse models [99]. Intratumoral injection provided long-term remission and most interestingly, also remission of subcutaneous tumors and bone metastases after intravenous administration.

## 4. Clinical Trials

Self-replicating RNA viruses have been subjected to a limited number of clinical trials (Table 3). In this context, alphaviruses have been subjected to few clinical trials, so far. For instance, VEE particles expressing CMV gB or a PP65/IE1 fusion protein were applied for a randomized, double-blind phase I clinical trial in CMV seronegative individuals [102]. Intramuscular or subcutaneous administration was well-tolerated with no clinically important changes and direct IFN-γ ELISPOT responses to CMV antigens were detected in all 40 vaccinated subjects. Moreover, immunization elicited neutralizing antibody and multifunctional T cell responses against all three CMV antigens. In another study, healthy HIV-negative volunteers were subjected to double-blind, randomized, placebo-controlled phase I trials in the United States and South Africa applying VEE expressing a nonmyristoylated form of Gag [103]. Subcutaneous administration of VEE-Gag was well-tolerated, but exhibited only modest local immune responses with low levels of binding antibodies and T cell responses. Although five serious adverse events were reported none were considered to be related to the administered vaccine. VEE particles capable of efficiently infecting DCs were employed for the expression of CEA in a clinical trial in patients with advanced cancer [104]. Intramuscular doses of 4 × 10^7^ IU to 4 × 10^8^ IU of VEE-CEA particles were given every three weeks for four immunizations. Repeated immunization induced clinically relevant CEA-specific T cell and antibody responses. The antibody-dependent cellular toxicity against tumor cells from human colorectal cancer metastases were mediated by CEA-specific antibodies. Moreover, longer overall survival was observed in patients with CEA-specific T cell responses. Propagation-defective VEE particles expressing the PSMA have also been subjected to a phase I clinical trial in patients with castration resistant metastatic prostate cancer (CRPC) [105]. Five doses of either 0.9 × 10^7^ IU or 3.6 × 10^7^ IU of VEE-PSMA were administered to patients with CRPC metastatic to the bone. Vaccinations were well-tolerated at both doses although only weak PSMA-specific signals were detected. Although neither clinical benefit nor robust immune responses were achieved, the elicited neutralizing antibodies suggest that dosing was suboptimal. In another approach, SFV particles expressing IL-12 (LipoVIL12) were encapsulated in liposomes, which provided passive tumor targeting and protection against host immune recognition [106]. LipoVIL12 was intravenously administered to melanoma and kidney carcinoma patients in a phase I clinical trial. Patients receiving LipoVIL12 showed transient (five days) up to 10-fold increased IL-12 plasma levels. The encapsulation enhanced tumor targeting and prevented host immune recognition after repeated injections. Furthermore, no toxicity was related to the treatment and the maximum tolerated dose (MTD) was determined as 3 × 10^9^ particles per m^2^.

Several VSV-based clinical trials have been conducted [107]. Two placebo-controlled, double-blind, dose-escalation phase I trials have been performed with recombinant VSV particles expressing the glycoprotein of a Zaire strain of Ebola virus [108]. A total of 78 volunteers received one of three doses (3 × 10^6^, 2 × 10^7^ or 1 × 10^8^ pfu) of VSV-ZEBOV to assess safety and immunogenicity of the vaccination. Some adverse event such as injection-site pain, fatigue, myalgia, and headache occurred. Lower titers were observed at day 28 for the dose of 3 × 10^6^ pfu in comparison to the other two doses.

Furthermore, a second dose at day 28 significantly increased the antibody titers at day 56, but the effect disappeared after 6 months. In another randomized, dose-ranging, observer-blind, placebo-controlled phase I trial, 40 participants received the attenuated VSVΔG-ZEBOV-GP vaccine [109]. No serious adverse events were encountered. All vaccinees developed immune responses comparable across all doses applied. Sustainable IgG titers were detectable throughout the whole study (180 days). Furthermore, another phase I study with VSV-ZEBOV showed good tolerance, no vaccine-related adverse events, and superior cellular immune responses and stronger interlocked cytokine networks for immunization with the highest dose of 2 × 10^7^ pfu [110]. In the Geneva phase I/II, dose-finding, placebo-controlled, double-blind study the VSV-ZEBOV dose was reduced to 3 × 10^5^ pfu compared to previous doses of 1–5 × 10^7^ pfu [111]. The lower dose improved tolerability, but decreased antibody responses. Moreover, it did not prevent vaccine-related arthritis, dermatitis, or vasculitis.

A randomized, placebo-controlled phase III trial was conducted in 1500 adults with the chimpanzee Ad3 (ChAd3-EBO-Z) and the recombinant VSV (rVSV∆G-ZEBOV-GP) vaccines in Liberia [112]. Adverse events including injection-site reactions, headache, fever, and fatigue occurred significantly more frequently in individuals receiving the active vaccine compared to placebo. Antibody responses were detected in 70.8% and 83.7% of subjects in the ChAd3-EBO-Z and the rVSV∆G-ZEBOV-GP groups, respectively, compared to 2.8% in the placebo group one month after vaccination. At 12 months the percentage for the ChAd3-EBO-Z and the rVSV∆G-ZEBOV-GP groups was 63.5% and 79.5%, respectively, with 6.8% in the placebo group. Another phase III trial was conducted in Guinea as an open-label, cluster-randomized ring vaccination study in suspected cases of Ebola virus disease (EBV) [113]. A total number of 7651 individuals were included, of which 4123 persons were assigned for immediate vaccination with rVSV-ZEBOV and 3528 persons assigned for delayed vaccination. No cases of EBV were detected in the immediate vaccination group after 10 days, whereas 16 cases of EBV were registered in the delayed vaccination group. No new cases of EBV were diagnosed in either group. Overall, the rVSV-ZEBOV vaccine was confirmed to be safe and showed promise as highly efficient in preventing EBV. Another phase III study was conducted in Guinea and Sierra Leone applying a single intramuscular vaccination with 2 × 10^7^ pfu of rVSV-ZEBOV [114]. In the randomized trial, 2119 individuals were immediately vaccinated and 2041 persons were vaccinated after a delay of 21 days after randomization. Vaccinated individuals were followed up for 84 days offering substantial protection against EBV with no cases of EBV discovered from day 10 after vaccination. Moreover, an individually-controlled phase II/III trial was conducted on health care and frontline workers in the five most EBV affected districts in Sierra Leone [115]. A single intramuscular dose was administered at enrollment or 18–24 weeks after enrollment. The outcome indicated that no EBV cases and no vaccine-related serious adverse events were reported. Finally, a randomized, double-blind, multicenter phase III trial was conducted in the United States, Spain, and Canada. [116]. Vaccination was taken place with doses of 2 × 10^7^ pfu and 1 × 10^8^ pfu of rVSV∆G-ZEBOV-GP and placebo for the assessment of safety and immunogenicity. The vaccine was generally well-tolerated. Although systemic adverse events occurred in comparison to placebo, no vaccine-related severe adverse events or deaths were reported. Overall, the results confirmed the safety of vaccination of the EBV risk population with rVSV∆G-ZEBOV-GP.

MV-Edm vaccine strains have been tested in clinical trials against breast, ovarian, head and neck cancer, glioblastoma, and myeloma [121,122]. In this context, an open-label, nonrandomized dose-escalation phase I trial was conducted with an unmodified vaccine strain MV-Edm Zagreb (MV-EZ) in patients with cutaneous T cell lymphomas [117]. Intratumor injections of MV-EZ on days 4 and 17 were preceded by subcutaneous IFNα injections (72 and 24 h prior to MV-EZ). The maximum tolerated dose was defined as 10^3^ TCID_50_. Complete regression of CTCL lesions was observed in one patient, while partial regression was observed in the other patients. Related to recombinant MV, a phase I trial was conducted in patients with advanced ovarian cancer by intraperitoneal injection of MV-CEA [118,123]. Administration of MV-CEA at doses of 10^3^ to 10^9^ TCID_50_ confirmed no dose-limiting toxicity. The best objective response comprised stable disease in 14 patients with a median duration of 88 days and a range of 55 to 277 days. All individuals vaccinated with higher dose levels (10^7^–10^9^ TCID_50_) accomplished stable disease, whereas only five out of 12 patients vaccinated with 10^3^–10^6^ TCID_50_) achieved it. MV-CEA has also been planned for a phase I clinical trial in patients with recurrent glioblastoma multiforme [119]. The study aims at treatment with a starting dose of 1 × 10^5^ TCID_50_ of MV-CEA escalating to the maximum dose level of 2 × 10^7^ TCID_50_. One group of patients will receive direct injections into the resection cavity and in the other group MV-CEA will be administered into recurrent tumors. So far, three patients have received 1 × 10^5^ TCID50 and three other patients 1 × 10^6^ TCID_50_ in the resection cavity showing no dose-limiting toxicity. Oncolytic MV vectors expressing the human sodium iodide symporter (NIS) have been subjected to a phase I trial [120]. Patients with relapsed refractory myeloma received intravenous MV-NIS or cyclophosphamide two days prior to MV-NIS treatment. The initial dose-escalation study (1 × 10^6^–1 × 10^9^ TCID_50_) revealed that the MTD was not reached. Therefore, doses of 1 × 10^10^ and 1 × 10^11^ TCID_50_ were tested and the latter dose was planned to be used in a phase II trial. A complete response was observed in one patient treated with 1 × 10^11^ TCID_50_. The response persisted for 9 months after which an isolated relapse occurred in the skull without recurrent marrow involvement. Irradiation of the lesion resulted in the patient remaining disease-free for an additional 19 months. Another patient had subjective softening and shrinking of her extramedullary plasmacytomas of her back and thighs.

## 5. Conclusions and Future Aspects

In summary, numerous preclinical and clinical studies have confirmed the feasibility of the approach of applying self-replicating RNA viruses for both preventive and therapeutic use for various diseases (Table 1, Table 2 and Table 3). In this context, immunization with self-replicating RNA viruses has generated strong immune responses and in many cases provided protection against challenges with lethal doses of infectious agents. Moreover, administration of self-replicating RNA viral vectors expressing anticancer, toxic and/or immunostimulatory genes have demonstrated tumor growth inhibition, regression and even complete tumor eradication, which has supported significant prolongation of survival profiles. Immunization has also provided prophylactic protection against challenges with tumor cells in animal models. One interesting aspect of applying self-replicating RNA viruses comprises of the possibility of using RNA replicons, replication-deficient and -competent particles, and layered DNA/RNA vectors. It provides certain flexibility in choosing the means of delivery vehicle for specific applications. Moreover, several attempts have been made to engineer vectors specifically targeting, replicating and killing tumor cells. One approach has been to apply oncolytic viral vectors, which has provided specific killing of tumor cells without affecting normal cells [124]. In another approach, recombinant SFV particles were encapsulated in liposomes, which provided passive targeting of tumors and protection against recognition by the host immune system [106]. Moreover, the limitation of vector use due to host immune responses has been addressed by engineering a polymer-coated MV-NPL vector based on the MV Edmonston strain with the N, P, and L genes of the wild-type MV strain [125]. The polymer-coated MV-NPL showed superior oncolytic activity in vitro compared to naked MV-NPL. Moreover, polymer-coated MV-NPL provides higher complement-dependent cytotoxicity and antitumor activities than naked virus in mice. In the context of optimization of immune responses, specific targeting of DCs has proven a useful approach demonstrating enhanced immune responses from VEE vectors transducing DCs [72]. Recently, the delivery to DCs and translation of replicon RNA from classical swine fever virus (CSFV) encoding influenza virus NP, belonging to flaviviruses, was improved by lipid formulations, which was demonstrated both in vitro and in vivo by induced immune responses against influenza NP [126]. Moreover, potential enhanced therapeutic efficacy has been addressed by various applications of combination therapy. For instance, a triple treatment combination of sunitinib, low-dose irradiation, and SFV-HPV E6,7 particles rendered mice tumor-free [79]. Similarly, combining VSV immunization with ruxolitinib administration enhanced responses in both subcutaneous and orthotropic xenograft models [89]. Furthermore, ruxolitinib and Polybrene or DEAE-dextran rendered VSV-resistant cells susceptible, which should aid VSV-based therapy [93].

Although a relatively small number of clinical trials have been conducted with self-replicating RNA viruses, there has been some promising results. Especially, several phase III trials on MV-based vaccines against EBV have provided good safety profiles and protection [112,113,114,115,116]. Alphavirus vectors have been subjected to clinical trials on infectious diseases [102,103]. So far, elicited immune responses have been relatively modest, which at least to some extent has been related to lack of dose optimization. In the context of using self-replicating RNA viruses for cancer therapy, less progress has been seen compared to infectious diseases. However, VEE-CEA particles showed prolonged survival in a phase I trial in pancreatic cancer. Moreover, promising results were obtained for liposome encapsulated SFV particles (LipoVIL12) in terminally ill melanoma and kidney carcinoma patients [106]. Also, MV vectors have shown regression of lymphoma lesions, stable disease in treatment of ovarian cancer [118], and complete response in one myeloma patient [120].

One important issue related to the utilization of any delivery system is safety. In the first phase, it is essential to provide high safety standards during laboratory research and large-scale production to ensure the protection of personnel. Related to self-replicating RNA viruses, special attention has been paid to the engineering of helper virus vectors used for virus preparation both at laboratory and large-scale. Initially, introduction of point mutations in the p62 precursor of the SFV E2 and E3 envelope genes rendered generated recombinant particles conditionally infectious requiring an additional activation step with α-chymotrypsin [127]. This second generation pSFV-Helper2 vector reduced the generation of replication-competent SFV particles to undetectable levels. Introduction of split helper systems for both SFV [128] and SIN [129] in which the capsid and envelope proteins are placed on separate plasmids generating high-titer particles eliminated production of recombinant-proficient alphavirus particles. Moreover, self-replicating RNA viruses—including alphaviruses, flaviviruses, MVs, and rhabdoviruses—have been classified at laboratory biosafety level 2 although the gene of interest expressed from the vector might impact the level [130]. Related to toxicity and adverse events observed in patients subjected to viral injections, VEE particles were well-tolerated, showed only local reactogenicity, and no clinically important changes [102]. However, although five serious adverse events were recorded in a phase I study in healthy HIV-uninfected individuals none were considered related to the vaccine [103]. The only adverse events related to immunization of Ebola patients with VSV vectors comprised of pain at the injection site, fatigue, myalgia, and headache [108]. Similarly, immunization of healthy volunteers with the VSVΔG-ZEBOV-GP vaccine showed only mild to moderate self-limited adverse events and injection-site pain and headache during a 14-day follow-up period [109]. Related to toxicity issues, a phase I study in CRPC patients demonstrated that VEE-PSMA administration was well-tolerated and no toxicity was observed [105]. Likewise, was well-tolerated in ovarian cancer patients showing no dose-limiting toxicity, MV-CEA [123]. Interestingly, a phase I trial in pancreatic cancer patients showed the feasibility of repeated injections [104]. Moreover, liposome-encapsulated SFV-IL12 particles could be repeatedly administered to kidney carcinoma and melanoma patients without demonstrating any toxicity, or virus- or liposome-related immune responses [106].

Looking into the future, continuous vector development aiming at delivery and safety improvements will certainly support the progress in therapeutic applications of self-replicating RNA viral vectors. Moreover, dose optimization studies, especially at the clinical level, needs to be conducted. As vaccine development and gene therapy approaches have taken giants leaps recently with classical approaches and more pioneering efforts using viral vectors and nucleic acids. The attractive features of self-replicating RNA viruses relate to the easy of virus production, broad host range, high safety levels due to no risk of chromosomal integration, targeting of DCs, but most importantly the extreme RNA replication in the cytoplasm, which supports high level transgene expression as the basis for generating strong immune responses. Today, RNA-based delivery provides an attractive approach, especially combined with either polymer- or liposome-based encapsulation strategies.

## Figures and Tables

**Figure 1 molecules-23-03310-f001:**
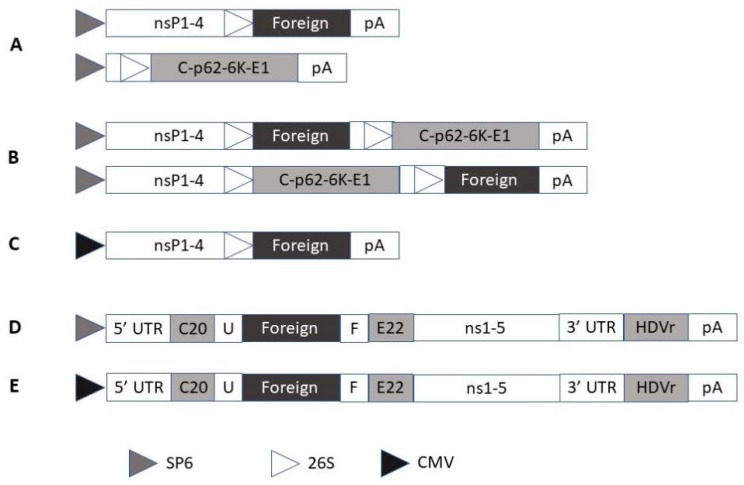
Self-replicating alphavirus and flavivirus vectors. (**A**) Replication-deficient alphavirus system with expression vector (above) and helper vector (below). (**B**) Replication-proficient alphavirus system including two alternative insertion sites. (**C**) DNA/RNA layered Semliki Forest virus (SFV) vector. (**D**, **E**) KUN vectors with SP6 and CMV promoters, respectively. 26S, subgenomic promoter; C20, first 20 amino acids of KUN C protein; CMV, cytomegalovirus; E22, last 22 amino acids of KUN E protein; F, Foot-and-mouth disease virus 2A autoprotease; HDVr, Hepatitis delta virus ribozyme; pA, polyadenylation signal; SP6, bacteriophage RNA polymerase; T7, phage T7 RNA polymerase promoter; U, mouse ubiquitin sequence; 3′ UTR, 3′ untranslated region; 5′ UTR, 5′ untranslated region.

**Figure 2 molecules-23-03310-f002:**
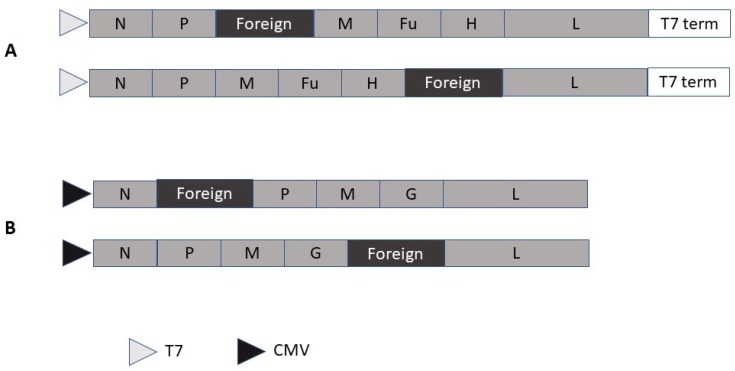
Self-replicating rhabdovirus and measles virus vectors. (**A**) Rabies virus vector with alternative insertion sites for foreign genes. (**B**) Measles virus vector with alternative insertion sites for foreign genes. CMV, cytomegalovirus; Fu, Measles virus fusion protein; G, Measles virus G protein; H, Measles virus hemagglutinin; L, Rabies or measles virus L protein; M, Rabies or measles virus matrix protein; N: Rabies or measles virus nucleocapsid protein; P, Rabies P or measles virus phosphoprotein; T7, phage T7 RNA polymerase promoter; T7 term, phage T7 terminator sequence.

**Table 1 molecules-23-03310-t001:** Examples of preclinical immunization with self-replicating RNA viruses targeting viral diseases.

Indication	Target/Antigen	Vector	Response	Ref.
Dengue	DV2	MV	Neutralizing Abs	[37]
	DV1-4	MV	Protection	[38]
Ebola	GP	KUN VLPs	Protection	[39]
	GP	VSV	Protection	[40,41]
	GP, NP	VEE VLPs	Protection	[42,43]
HBV	HBsAg	MV	Humoral Abs	[45]
	MHB	SFV-VLVs	Protection	[46]
HIV	Env	SFV VLPs	Humoral response	[47]
	Env	SFV RNA	Antibody response	[48]
	Env/Gag/Pol	SFVVLPs/RNA	Ag-specific immune response	[49]
	Env/GagPolNef	SFV DNA	T cell and IgG responses	[50]
Influenza	NP	SFV VLPs	Mucosal immune response	[51]
	HA	VEE VLPs	Protection	[52]
	HA, NP	SFV VLPs	Protection	[53]
	HA	VEE RNA	Protection	[54]
Lassa	Glycoprotein	VSV VLPs	Protection	[55]
	GPC	VEE VLPs	Protection	[56]
	Lassa	Lassa VLPs	Protection	[57]
MERS-CoV	MERS-S	MV	Protection	[58]
SARS-CoV	SARS-CoV GP	VEE VLPs	Protection	[59]
MPV	MPV-F	VEE VLPs	Reduced viral load	[60]
RSV	RSFV-F	SFV-VLPs	Protection	[53]
	RSV-F	VEE VLPs	Reduced viral load	[60]
	RSV-F	VEE LNPs	Protection	[61]
SIV	Gag-pol	KUN VLPs	Protection	[62]
	SIVmacJ5	SFV VLPs	Cytotoxic T cell response	[63]
TBEV	GP	SFV VLPs	Protection	[53]
VEE	VEE replicon	VEE miRNA	VEE inhibition	[64]
	VEE RdRp	VEE miRNA	VEE inhibition	[65]

Ag, antigen; DV, Dengue virus envelope protein; GCP, glycoprotein; GP, glycoprotein; HA, hemagglutinin; HBV, hepatitis B virus; HBsAg, hepatitis B surface antigen; HIV, human immunodeficiency virus; KUN, Kunjin virus; LNPs, lipid nanoparticles; MERS-CoV, Middle East respiratory syndrome coronavirus; MERS-S, Middle East respiratory syndrome spike protein; MHB, HBV middle surface envelope glycoprotein; miRNA, micro RNA; MV, measles virus; NP, nucleoprotein; RdRp, RNA-dependent RNA polymerase; RSV, respiratory syncytial virus; SARS-CoV, severe acute respiratory syndrome corona virus; SFV, Semliki Forest virus; SIV. Simian immunodeficiency virus; TBEV, Tick-borne encephalitis virus; VEE, Venezuelan equine encephalitis virus; VLPs, virus-like particles; VLVs, virus-like vesicles; VSV, vesicular stomatitis virus.

**Table 2 molecules-23-03310-t002:** Examples of preclinical immunization studies of self-replicating RNA viruses targeting cancers.

Cancer	Target/Antigen	Vector	Response	Ref.
Brain	SLAM, EGFR	MV	Tumor regression	[66]
	Endostatin	SFV VLPs	Tumor inhibition	[67]
	miR124	SFV VLPs	Prolonged survival	[68]
Breast	CEA	MV	Prolonged survival	[69]
	HER2/neu	SIN DNA	Prolonged survival	[70]
	Δneu	VEE + DCs	Tumor regression	[71]
	VEGFR-2	SFV VLPs	Tumor inhibition	[72]
Colon	VEGFR-2	SFV VLPs	Tumor inhibition	[72]
	G-CSF	KUN VLPs	Tumor regression	[73]
	LacZ	SFV RNA	Prolonged survival	[74]
Cervical	HPV16 E7	VEE VLPs	Tumor prevention	[75]
	HPV E6,7	SFV VLPs	Tumor eradication	[76]
	HPV E6, E7	SFV DNA	Tumor eradication	[77]
	HPV E6,7	SFV + I	>Antitumor activity	[78]
	HPV E6,7	SFV + I + Sun	Tumor-free mice	[79]
Lung	Dengue	Dengue	Lung susceptibility	[80]
	EGFP	SFV VLPs	Tumor regression	[81]
	EGFP	SFV(VA7)	Prolonged survival	[82]
	SLAM	MV	Tumor suppression	[83]
Melanoma	G-CSF	KUN VLPs	Tumor regression	[73]
	SIINFEKL	YFV	Tumor regression	[84]
	TRP-2	VEE	Prolonged survival	[85]
	TRP-2	VEE + mAbs	Tumor regression	[86]
	VEGFR2, IL12	SFV DNA	Tumor regression	[87]
	Surv, β-hCG	Combination	Prolonged survival	
	VSV	VSV-GP	Prolonged survival	[88]
Ovarian	VSV	VSV-GP + Rux	Oncolytic activity	[89]
	IL12	SIN + CPT-11	Long-term survival	[90]
	OVA	SFV + VV	Antitumor response	[91]
Pancreatic	VSV	VSV	PDA susceptibility	[92]
	VSV	VSV + Rux	HPAF-II susceptible	[93]
	SLAM	MV	Tumor suppression	[94]
Prostate	CEA	MV	Prolonged survival	[95]
	PSMA	VEE	Prolonged survival	[96]
	STEAP	VEE	Prolonged survival	[97]
	PSCA	VEE	Prolonged survival	[98]
	VSV	VSV-GP	Long-term remission	[99]

CEA, carcinoembryonic antigen; CPT-11, irinotecan; EGFP, enhanced green fluorescent protein; EGFR, epidermal growth factor receptor; G-CSF, granulocyte colony-stimulating factor; HPV, human papilloma virus; I, irradiation; KUN, Kunjin virus; miRNA, micro RNA; mAbs, monoclonal antibodies; MV, measles virus OVA, ovalbumin; PSCA, prostate stem cell antigen; PSMA, prostate-specific membrane antigen; Rux, ruxolitinib; SFV, Semliki Forest virus; SIN, Sindbis virus; SIINFEKL, chicken ovalbumin epitope; SLAM, signaling lymphocyte activation molecule; STEAP, six-transmembrane epithelial antigen of the prostate; Sun, sunitinib; TRP-2, tyrosine-related protein-2; VEE, Venezuelan equine encephalitis virus; VEGFR-2, vascular endothelial growth factor receptor-2; VLPs, virus-like particles; VSV, vesicular stomatitis virus; VV, vaccinia virus.

**Table 3 molecules-23-03310-t003:** Examples of clinical trials applying self-replicating RNA viruses.

Disease	Vector/Antigen	Phase	Response	Ref.
CMV	VEE-gB/pp65	Phase I	CMV-spec Abs	[102]
HIV	VEE-Gag	Phase I	Low level Ab responses	[103]
Ebola	VSV-ZEBOV	Phase I	Safe, Ab responses	[108]
	VSVΔG-ZEBOV	Phase I	Safe, sustainable Ab titers	[109]
	VSV-ZEBOV	Phase I	Safe, cellular immune responses	[110]
	VSV-ZEBOV	Phase I/II	Reduced dose, better tolerability	[111]
	VSVΔG-ZEBOV	Phase III	Ab responses	[112]
	VSV-ZEBOV	Phase III	Safe, efficient EBV prevention	[113]
	VSV-ZEBOV	Phase III	Safe, substantial EBV protection	[114]
	VSV-ZEBOV	Phase II/III	Safe, no EBV cases, no SAEs	[115]
	VSVΔG-ZEBOV	Phase III	No EBV related SAEs	[116]
Pancreatic CA	VEE-CEA	Phase I	CEA-spec Abs; prolonged survival	[104]
CRPC	VEE-PSMA	Phase I	Neutralizing Abs	[105]
CTCL	MV-EZ	Phase I	Regression of CTCL lesions	[117]
Melanoma	LipoVIL12	Phase I	Safe tumor targeting	[106]
Kidney CA	LipoVIL12	Phase I	Safe tumor targeting	[106]
Ovarian CA	MV-CEA	Phase I	Stable disease	[118]
Glioblastoma	MV-CEA	Phase I	No dose-limiting toxicity	[119]
Myeloma	MV-NIS	Phase I	Complete response in one patient	[120]

Abs, antibodies; CA, cancer; CEA, carcinoembryonic antigen; CMV, cytomegalovirus; CRPC, castration resistant metastatic prostate cancer; CTCL, cutaneous T cell lymphoma; EBV, Ebola virus disease; HIV, human immunodeficiency virus; LipoVIL12, liposome encapsulated SFV-IL21 particles; MV, measles virus; MV-EZ, measles virus Edmonston Zagreb; NIS, sodium iodide symporter; PSMA, prostate-specific membrane antigen; SEAs, serious adverse events; SFV, Semliki Forest virus; VEE, Venezuelan equine encephalitis virus; VSV, vesicular stomatitis virus; ZEBOV, glycoprotein of Zaire Ebola virus.

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
