# Peer review of "Self-Replicating RNA Viruses for RNA Therapeutics"

_molecules, 2018, doi:10.3390/molecules23123310_

Reviewer 1 Report

            This manuscript reviews the applications for self-replicating RNA viruses and corresponding pre-clinical studies and clinical trials of viral diseases and cancer treatment. Those studies assessed the feasibility, safety, efficiency of applying this novel self-replicating RNA viruses for both preventive and therapeutic purpose, which provides the foundation of continuous vector development for delivery and safety improvements. Overall, the review is timely important, comprehensive, and detailed. However, there are some parts that need to be addressed to further improve the quality of this work:

1.      In the abstract, the purpose and subject of this review should be better identified.

2.      In the introduction, please discuss the example of the very first RNA interference therapeutic agent approved by FDA and cite the corresponding work (The New England journal of medicine, 379, 11-21, 2018)

3.      The schematic that explains the therapeutic mechanisms of action for the self-replicating RNA viruses should be added as Figure 1.

4.      This review very briefly discussed using lipid nanoparticles, while the delivery problem is very important. More delivery methods and comparing their delivery efficiency is necessary to be discussed.

5.      There are several places mentioning the “immune response”. What kind of immune response is normally observed for these technologies? Which PRRs are mostly involved? What are the pathways and mechanisms that get activated? The inclusion of some graphs or tables to “summarize” the effects is suggested.

Author Response

Q: The purpose and subject of this review should be better identified

R: The text has been added to the abstract.

Q; Please discuss the example of the very first RNA interference therapeutic agent approved by FDA

R: The reference is incorrect, but a short description of Patisiran is added to the introduction.

Q: The schematic that explains the therapeutic mechanisms of action for the self-replicating RNA viruses should be added as Figure 1.

R: The strategy is fairly clear applying self-replicating RNA virus vectors to induce immune responses, then to be measured by antibody titers and ultimately testing protection. If the reviewer/editor requests to include a figure I will be happy to do it.

Q:This review very briefly discussed using lipid nanoparticles, while the delivery problem is very important. More delivery methods and comparing their delivery efficiency is necessary to be discussed.

R:I agree that delivery is an important issue. However, this review focuses on RNA viruses and I feel it would be a too big expansion to include nanoparticle-based delivery systems.

Q: There are several places mentioning the “immune response”. What kind of immune response is normally observed for these technologies? 

R: More specific description of immune responses has been added to Table 1.

Reviewer 2 Report

The manuscript by Kenneth Lundstrom provides a  comprehensive review of the current literature regarding  self-replicating RNA viruses and their use for therapeutic applications.  The manuscript is well structured and includes sections covering  a general overview of the field, type of viruses and discussion of  preclinical and clinical studies. It would be beneficial for the readers  if, in addition to the types of therapeutic application of these  viruses, the author also describes safety concerns  associated with their use. For example, the author may consider adding a  section called “Satefy Considerations” or “Safety Concerns” or “Safety  aspects” in which, he describes current occupational, environmental and  patient safety experience with therapeutic   self-replicating viruses. Were there any adverse immune effects  noticed in preclinical and clinical studies? If so, how were they  overcome? What types of toxicities were observed? What is needed to  ensure safety of people who work in production facilities  and labs working with these viruses? What impact they may have on the  environment? Some experience with injection site reactions was mentioned  in the clinical trials section of the manuscript. But I think the  manuscript may have a greater impact if the safety  is discussed more explicitely.  I recommend this manuscript for  publication after this point is addressed along with several minor edits  listed below:

Page  3, line 109 – “studies targeting both infectious diseases (Table 1)”.  It appears that a continuation is missing “Both infectious and xxxxxx  diseases”. Alternatively,  word “both”  needs to be deleted

Page 4, line 142 – please correct a typo in word “self-replicating” (currently spelled “selkf-replicating”)

Page 4, line 149 – please correct a typo in word “than” (currently spelled “tan”)

Page 5, line 171 – please correct a typo in word “from” (currently spelled “ffrom’)

Page 5, line 175 – please correct a typo in “challenged” (currently spelled “challengesd”)

Page 10, line 357 – please correct a typo in “was” ( currently spelled “weas”)

Page 14, line 525 – please correct a typo in “specifically” (currently spelled specically”)

Please  check one more time the entire manuscript to capture and correct any  other possible typos, which this reviewer may have missed

Author Response

Q: For example, the author may consider adding a section called “Safety Considerations” or “Safety Concerns” or “Safety aspects”.

R: A paragraph has been added on safety issues.

Q: Page 3, line 109 – “studies targeting both infectious diseases (Table 1)”.  It appears that a continuation is missing “Both infectious and xxxxxx  diseases”. Alternatively, word “both”  needs to be deleted

Page 4, line 142 – please correct a typo in word “self-replicating” (currently spelled “selkf-replicating”)

Page 4, line 149 – please correct a typo in word “than” (currently spelled “tan”)

Page 5, line 171 – please correct a typo in word “from” (currently spelled “ffrom’)

Page 5, line 175 – please correct a typo in “challenged” (currently spelled “challengesd”)

Page 10, line 357 – please correct a typo in “was” ( currently spelled “weas”)

Page 14, line 525 – please correct a typo in “specifically” (currently spelled specically”)

Please check one more time the entire manuscript to capture and correct any  other possible typos, which this reviewer may have missed

R: All specific corrections addressed by the reviewer have been made and manuscript checked for typos.